# Fetal Oxygenation from the 23rd to the 36th Week of Gestation Evaluated through the Umbilical Cord Blood Gas Analysis

**DOI:** 10.3390/ijms241512487

**Published:** 2023-08-06

**Authors:** Luca Filippi, Francesca Pascarella, Alessandro Pini, Maurizio Cammalleri, Paola Bagnoli, Riccardo Morganti, Francesca Innocenti, Nicola Castagnini, Alice Melosi, Rosa Teresa Scaramuzzo

**Affiliations:** 1Department of Clinical and Experimental Medicine, University of Pisa, 56126 Pisa, Italy; 2Neonatology Unit, Azienda Ospedaliero-Universitaria Pisana, 56126 Pisa, Italy; fro221295@gmail.com (F.P.); f.innocenti17@gmail.com (F.I.); n.castagnini@studenti.unipi.it (N.C.); alicemelosi@icloud.com (A.M.); r.scaramuzzo@ao-pisa.toscana.it (R.T.S.); 3Department of Experimental and Clinical Medicine, University of Florence, 50121 Florence, Italy; 4Unit of General Physiology, Department of Biology, University of Pisa, 56126 Pisa, Italy; maurizio.cammalleri@unipi.it (M.C.); paola.bagnoli@unipi.it (P.B.); 5Section of Statistics, Azienda Ospedaliero-Universitaria Pisana, 56126 Pisa, Italy; riccardo-morganti@ao-pisa.toscana.it

**Keywords:** newborn, intrauterine hypoxia, fetal hypoxia, differentiation

## Abstract

The embryo and fetus grow in a hypoxic environment. Intrauterine oxygen levels fluctuate throughout the pregnancy, allowing the oxygen to modulate apparently contradictory functions, such as the expansion of stemness but also differentiation. We have recently demonstrated that in the last weeks of pregnancy, oxygenation progressively increases, but the trend of oxygen levels during the previous weeks remains to be clarified. In the present retrospective study, umbilical venous and arterial oxygen levels, fetal oxygen extraction, oxygen content, CO_2_, and lactate were evaluated in a cohort of healthy newborns with gestational age < 37 weeks. A progressive decrease in pO_2_ levels associated with a concomitant increase in pCO_2_ and reduction in pH has been observed starting from the 23rd week until approximately the 33–34th week of gestation. Over this period, despite the increased hypoxemia, oxygen content remains stable thanks to increasing hemoglobin concentration, which allows the fetus to become more hypoxemic but not more hypoxic. Starting from the 33–34th week, fetal oxygenation increases and ideally continues following the trend recently described in term fetuses. The present study confirms that oxygenation during intrauterine life continues to vary even after placenta development, showing a clear biphasic trend. Fetuses, in fact, from mid-gestation to near-term, become progressively more hypoxemic. However, starting from the 33–34th week, oxygenation progressively increases until birth. In this regard, our data suggest that the placenta is the hub that ensures this variable oxygen availability to the fetus, and we speculate that this biphasic trend is functional for the promotion, in specific tissues and at specific times, of stemness and intrauterine differentiation.

## 1. Introduction

The embryo and the fetus live in a physiologically hypoxic environment (the so-called Everest in utero) [1], although the intrauterine environment is characterized by a variable level of hypoxia throughout pregnancy.

During the first stages of pregnancy, the concentrations of oxygen are very low, similar to those measurable within the non-pregnant uterus [2,3]. Over the first weeks of pregnancy, placental development favors the increase in oxygen availability to the fetoplacental unit, and, at the beginning of the second trimester of gestation, placental oxygenation triples the oxygen availability, reaching a maximum around the 16th week of gestation [3]. From this week onwards, a slow, gradual reduction in placental oxygen levels is observed, and this trend appears to be consistent with a progressive reduction in fetal oxygenation status [3]. In fact, a series of studies carried out by analyzing human umbilical cord venous and arterial blood samples obtained via cordocentesis demonstrated decreasing fetal partial pressure of oxygen (pO_2_) and saturation of oxygen (SaO_2_), which start from the 16–18th week of gestation and progress with advancing gestation [4,5,6,7].

However, some considerations led us to imagine a reversion of the trend of oxygenation during the final stages of pregnancy. This hypothesis was inspired by the observation that in many animals, the vasculature is still immature at birth and that its maturation occurs after birth. This is particularly evident in the brain of rodents, whose appropriate glial–vascular interaction is instituted after birth [8,9] and in the retina of mice, which is avascular at birth but begins to vascularize during the first week after birth [10] or even later [11], demonstrating, at least chronologically, a strict relationship between vessel maturation and an increase in oxygen exposure. If vascular maturity requires an increase in oxygen levels, it is legitimate to imagine that in humans, whose vascularization is completed during the last weeks of intrauterine life [12], there is a progressive increase in oxygen tension in the more advanced stages of pregnancy.

To verify if oxygenation increased in the last weeks of pregnancy, we recently performed an observational study that assessed the umbilical gas analysis collected from a cohort of healthy newborns with gestational ages ≥ 37 weeks. Results demonstrated a progressive increase in fetal oxygenation from the 37th to the 41st weeks of gestation (approximately an increase of about 1 mmHg per week) [13]. These findings are in accordance with a previous study that reported a slight increase in cord venous pO_2_ between near-term and term newborns [14]. The pooled analysis of studies evaluating fetal oxygen levels suggests that oxygenation during intrauterine life continues to vary after placenta development, taking on a biphasic trend. Fetuses, in fact, from mid-gestation to near-term grow in an environment progressively more hypoxic, while, approaching the term of pregnancy, oxygenation reverses its trend. Knowledge of this dynamic assumes a particular importance because it helps us understand the physiological mechanisms that guarantee the maturation of vascularization, whether it takes place after birth, as in rodents, or in utero, as in humans.

The main objective of this study is to reconstruct, with the same methodology used to analyze the blood gas variations in term infants, the fetal oxygenation status from the 23rd to the 36th weeks of gestation. Our objectives are to verify that healthy fetuses become more hypoxemic over the weeks, to confirm that this trend at a certain point is reverted, and to identify the time when this reversal occurs.

## 2. Results

Out of 12,544 newborns born between 2016 and 2022, 11,375 newborns with GA ≥ 37 weeks were excluded from the analysis, and 1169 preterm newborns were eligible for this study. Umbilical cord gas analyses of 560 newborns were not performed, incomplete, or unreliable. Out of 609 samples, 5 with acidosis at birth were excluded (Figure 1).

Of the 604 newborns enrolled, males were prevalent (310/604, 51.2%). Neonates born by vaginal delivery were a minority (148/604, 24.5%). Table 1 shows the cumulative demographic and gas analytical parameters of all newborns.

According to what was recently observed in term newborns [13] and also in preterm neonates, the modality of delivery influences the blood gas analysis. Neonates born by vaginal delivery showed higher values of pO_2_ and lactate both in venous and arterial cord samples. Lower levels of pH, partial pressure of carbon dioxide (pCO_2_), and bicarbonate were evident only in venous cord samples. Neonates born by cesarean section displayed an oxygen extraction significantly higher than neonates born by spontaneous delivery, in line with what was demonstrated in term neonates [13]. To verify whether the incidence of vaginal or caesarean delivery substantially changed as gestation progressed, data from blood gas analysis were arbitrarily grouped into six newborn groups: a group of preterm newborns from 23 to 26^+6^ weeks of gestation, 27–28^+6^, 29–30^+6^, 31–33^+6^, 34–35^+6^, and 36–36^+6^ (Figure 1). The percentage of vaginal or cesarean delivery was not statistically different among the groups, allowing us to evaluate the blood samples as a whole, regardless of the type of delivery, without further stratification.

To evaluate whether oxygen tension varied during pregnancy, data from venous and arterial pO_2_, as well as oxygen extraction, were assessed in relation to the progression of pregnancy (Figure 2). Values from umbilical venous samples revealed a biphasic trend of oxygen level, as suggested by the significance of the quadratic regression: pO_2_ levels progressively decrease until approximately 230–240 days of gestation (approximately the 33rd–34th week of gestation), when pO_2_ appears to reach its lowest point; then, starting from the 34th week of pregnancy, pO_2_ increases (Figure 2A), to then continue, ideally, with the positive progression observed from the 37th to the 41st weeks of gestation [13]. This tendency is better highlighted in Appendix A, where the data were stratified according to groups of weeks of pregnancy. The arterial pO_2_ level also appears to be linearly reduced with the progression of gestation (Figure 2B), with a minimal increase in the weeks near term (Appendix A). Overall, oxygen extraction did not show significant change throughout the period considered (Figure 2C), even though a modest increase was evident comparing the groups of 27–28^+6^ and 29–30^+6^ towards the more advanced stages of pregnancy (Appendix A).

The levels of venous and arterial hemoglobin (Hb) progressively increased from the 23rd week onwards, reaching a plateau approximately at 230–240 days of gestation (around weeks 33–34 of gestation) (Figure 3A,B). The Hb rise appears to follow a trend opposite to that followed by pO_2_, with an increase of around 1 g/dL per group of gestational age, as shown in Appendix A. This increase explains why, despite the reduced intake of oxygen from the placenta to the fetus, the venous content of oxygen remained stable throughout the period examined (Figure 3C), excluding a minimal increase during the weeks near term (Appendix A).

Simultaneously with the decrease in pO_2_, a linear reduction in venous and arterial pH was observed as pregnancy proceeded (Figure 4A,B), coupled with a mild reduction in venous base excess (Figure 4C), but not in arterial base excess (Figure 4D), and a mild increase in arterial bicarbonate (Figure 4F), but not in venous (Figure 4E). This tendency is emphasized in Appendix A, where the data were stratified according to groups of weeks of pregnancy.

In analogy to what was observed in term fetuses, also in preterm fetuses, blood gas analysis displayed for the umbilical cord content of CO_2_ an opposing trend if compared with the trend of oxygen, with a progressive surge both in venous (Figure 5A) and arterial samples (Figure 5B). This increase is progressive at least until the group with gestational age 31–33^+6^ weeks (Appendix A). Starting from the 34th week of pregnancy, pCO_2_ decreases and then continues, ideally, with the negative progression recently described in term fetuses [13]. Overall, CO_2_ production did not seem to change significantly (Figure 5C), even though, during the more advanced weeks of gestation, we observe an apparent increase in CO_2_ production (Appendix A) that then continues, ideally, with the progression recently described from the 37th to the 41st week of gestation [13].

Between the 23rd and 36th weeks of gestation, no significant change in lactate levels was observed either in venous samples (Figure 6A) or in arterial samples (Figure 6B). A mild increase in fetal lactate production seemed evident (Figure 6C) as a consequence of a mild increase detected in newborns belonging to more advanced gestational ages (Appendix A).

## 3. Discussion

Conception, embryogenesis, and fetal growth occur in the female reproductive tract, where the oxygen concentration is very low. A physiological hypoxia has been indicated as the key regulator of the harmonious processes of placental and embryonal development [15,16]. The importance of this peculiar environmental oxygenation is witnessed by the improved live birth rate of preimplantation embryos cultured under hypoxic conditions compared with normoxic cultures [17]. However, the embryo and the following fetoplacental unit grow and differentiate in a milieu where the levels of hypoxia undergo evident and recurring variations throughout the pregnancy.

Placental and embryonic cells react to a hypoxic environment with a series of adaptive adjustments to gene expression. The upregulation of hypoxia-inducible factors (HIFs) represents the hub through which hypoxia promotes placental development [18]. Under hypoxia, HIF-1α translocates into the nucleus and dimerizes with HIF-1β, leading to the transcription of several hundred specific target genes [19], favoring cell survival in a hypoxic environment (induction of specific enzymes involved in energy metabolism, erythropoiesis, and angiogenesis). [20]. Therefore, HIF modulation guarantees at the same time the proliferation and metabolic adaptation of embryonal cells, placental development, and trophoblast differentiation [21,22].

HIF regulation during pregnancy is complex. Although HIF is regulated also through oxygen-independent mechanisms such as hormones (angiotensin II), cytokines (interleukin-1β, tumor necrosis factor α, and NF-kβ), or growth factors (transforming growth factor-β1 and insulin-like growth factor), all of which are significantly upregulated during pregnancy [23], oxygen levels (and their fluctuations) represent the main regulator of placental HIF levels, placenta development, and normal mammalian embryo morphogenesis [24]. The pivotal role of HIF for embryonic/fetal wellness is indirectly confirmed by the high embryonic lethality, dysmorphogenesis, or severe placental defects observed in HIF-1α-/- mouse embryos [18].

Although hypoxia and HIF are indispensable for the harmonious development of the fetoplacental unit, from the first moments of pregnancy, it is evident that the levels of oxygen and HIF change following a precise dynamic.

Conception occurs when a sperm cell successfully fertilizes an egg cell in the fallopian tube, at an oxygen concentration of approximately 5–7% [25]. By this time, the morula reaches the uterine cavity, where the oxygen concentration is significantly reduced to 2% [26,27]. The transition to this more hypoxic environment is essential in the first 2–3 weeks of life to allow the proliferation, implantation, anchoring, and invasion of the blastocyst into the maternal uterus [28]. In fact, on one side, hypoxia (and the up-regulation of HIFs) is crucial to maintaining the pluripotent status of embryonic stem cells [29], as well as of cytotrophoblast cells, which are usually considered to be trophoblast stem cells [3], and to promoting their proliferation [30]. On the other side, exposure to a decreasing oxygen gradient during the travel from the oviduct to the uterus promotes a metabolic shift of cytotrophoblast cells from oxidative phosphorylation towards a glycolytic metabolism (Warburg effect) [31,32,33]. This metabolic adaptation confers a proliferative advantage since it promotes the production of many intermediates of the pentose phosphate pathway, such as ribose sugars necessary for nucleic acid synthesis, and induces the production of elevated lactic acid levels that are useful to promote nesting in surrounding tissues [34]. In fact, the increased production and extrusion of lactate facilitate the disaggregation of uterine tissues and promote trophoblast invasion [35].

Increasing hypoxia and high levels of HIF during the first weeks of life are also necessary to assure cytotrophoblast transmigration across the uterine epithelium and their differentiation into extravillous trophoblasts [18,36], which in turn migrate along the lumen of the spiral arterioles. These cells progressively remodel the spiral arteries, replacing the smooth muscle and the elastic lamina of the vessel. In this way, the low-flow, high-resistance original spiral arteries become high-flow, low-resistance vessels and assure, in the following weeks of pregnancy, a gradual increase in oxygen delivery [28].

In summary, the intrauterine low-oxygen tension promotes embryonic cytotrophoblast cell proliferation, their intrauterine invasion, and interconnections with the uterine spiral arterioles [37], thus laying the foundations for a subsequent increase in oxygen levels. In the meantime, HIF upregulation promotes the concomitant activation of a series of angiogenic factors, including vascular endothelial growth factor (VEGF), basic fibroblast growth factor, platelet-derived growth factor, angiopoietin-1, angiopoietin-2, Tie2, and monocyte chemoattractant protein-1 [38]. These proangiogenic factors in turn raise the foundation for embryonal vascular and placental development and a further increase in oxygen availability [3]. In turn, such an increase in microenvironmental oxygen promotes the differentiation of cytotrophoblasts into multinucleated syncytiotrophoblasts, which are necessary to secrete several hormones for pregnancy maintenance [39].

Essentially, what happens to the embryo in the first weeks of life represents a plastic demonstration that hypoxia is indispensable; however, within this hypoxic environment, we observe a fine regulation of oxygen levels, of HIF, and consequently of the expression of its target genes aimed at modulating apparently contradictory functions, such as the expansion of stemness and differentiation.

Placentation explains why placental oxygen tension, which is less than 20 mmHg (more or less 2% O_2_) during the first weeks of human gestation (approximately until the 10th week) [2,3], increases to around 60 mmHg (roughly 8% O_2_) at the beginning of the second trimester of gestation [2]. The increased oxygen availability from the first weeks to the early weeks of the second trimester justifies why the fetal habitat can be considered a dynamic environment where hypoxia initially attenuates.

While recent studies performed on healthy fetal animals reported no significant variations in blood gas and acid–base status during gestation [40,41,42,43,44,45,46,47], the available data for human fetuses obtained via cordocentesis, although limited, indicate a progressive reduction in fetal oxygen levels starting from the 16–18th weeks of gestation toward its end [4,5,6,7]. The reduced oxygen supply to the fetuses is explained by the increased placental oxygen extraction with advancing gestation, as demonstrated by samples of maternal blood drawn from the subchorial lake under the chorionic plate [4]. However, these studies were performed on pregnancies complicated by fetal anomalies, a variety of prenatal pathologies, or maternal infections [4,5,6,7]; thus, such results would need to be confirmed in a healthy population. The fluctuations of oxygen levels are further confirmed by the analysis of a large amount of umbilical cord blood gas analysis, in which a clear trend reversal of oxygenation in the last weeks of gestation was demonstrated [13,14]. Although the investigation of the mechanisms underlying the increased oxygenation was beyond the scope of these studies, this phenomenon has been attributed to increased oxygen diffusion from the placenta to the fetus, probably correlated with the physiological aging of the placenta. In fact, the thinning of the placental barrier surface area due to a reduction in the distance between fetal vessels and the trophoblastic membrane over time favors an increase in placental oxygen transport [48]. In conclusion, the literature shows data apparently contradictory; while, from mid-gestation to near term, several historical studies report a progressive reduction in fetal oxygenation [4,5,6,7], other more recent studies describe increased oxygenation during the last weeks of gestation [13,14]. This contradiction can only be explained by imagining a biphasic trend of fetal oxygenation, with a first phase characterized by a progressive reduction in fetal oxygenation, followed by a phase of increased oxygenation.

The results of the present study confirm that fetuses grow in a dynamic low-oxygen environment and demonstrate for the first time that the concentration of oxygen follows a biphasic trend. In fact, at least from the 23rd week onward, we observed a progressive and linear reduction in venous pO_2_ that slows and then reverses at around 33–34 weeks of gestation. A similar trend is also observed regarding the arterial samples, and therefore the oxygen extraction values do not change substantially during this time. These data suggest that the fetus is subject to a slow and progressive reduction in oxygen supply from the placenta, at least from the 23rd until the 33–34th week, which represents the inflection point between the two phases. This observation is in line with what has been observed from the second to the third trimester of pregnancy in the intervillous space of the human placentas, where mean oxygen tension measured gradually declines [3,4,49].

If we consider that the pO_2_ level in mothers remains stable throughout pregnancy [50] in concomitance with the uterine blood flow [51], the gradual reduction in venous umbilical pO_2_ suggests that placental growth and its increasing metabolic activity progressively make less oxygen available to the fetus. It is well known that placental oxygen consumption accounts for a large percentage of the collective fetal and placental oxygen consumption and that this percentage increases over time [15,51]. Therefore, our data suggest that the increasing consumption of oxygen by the placenta causes a progressive accentuation of fetal hypoxemia until the 33–34th week of pregnancy. In the face of a continuous reduction in the placental oxygen supply, the venous pCO_2_ progressively increases until the 33–34th week of gestation, confirming the hypothesis that these changes in the supplies to the fetus are consequent to an increased metabolic activity of the placenta. The combination of the reduction in oxygenation with the increase in pCO_2_ explains why the venous pH gradually decreases, at least up to the 33–34th week of pregnancy.

Increasing fetal hypoxemia is confirmed by the progressive increase in Hb concentration, the synthesis of which is known to be induced by low levels of oxygen through the activation of the HIF/erythropoietin axis [52]. The present study confirms that the Hb level gradually and progressively increases as pregnancy progresses, starting from the 23rd week of gestation, showing a trend opposite to that of pO_2_, as already reported by several authors [53]. Despite the progressive decrease in pO_2_, the increased Hb explains why the umbilical venous blood oxygen content remains unchanged. This strategy allows the fetus, at least from the 23rd to the 33–34th weeks of pregnancy, to simultaneously obtain two results: On the one hand, it guarantees a constant oxygen content and therefore does not deprive the peripheral tissues of the necessary oxygen, but at the same time, the concentration of oxygen dissolved in the blood is reduced. This explains why the fetus becomes progressively more hypoxemic despite the constant oxygen content that maintains the fetus’s stable hypoxia status. The maintenance, from the 23rd to the 33–34th week of gestation, of this efficient balance is evidenced by the absolute stability of lactate values.

However, from the 33–34th week forward, this balance gets broken and oxygenation starts to increase, as recently described in full-term newborns [13]: This gestational age represents the watershed beyond which umbilical venous oxygenation increases, probably due to a physiologic aging of the placenta that favors an increased diffusion of oxygen [48], and consequently, the increase in fetal Hb levels stops. The oxygen availability to fetuses during the last weeks of gestation is probably even more significant because, in late gestation, the percentage of fetal Hb decreases as adult Hb (which has a lower affinity for oxygen) begins to increase [54]. A comparable reversal is observed for venous pCO_2_, which increases until the 34th week; after that, pCO_2_ stops increasing, ideally continuing to decrease until the end of pregnancy [13]. The arterial pCO_2_ instead seems to continue to increase even after the 34th week, suggesting a possible increase in fetal pCO_2_ production and therefore a possible metabolic shift toward mitochondrial activation, which will be more evident after 37 weeks [13].

Therefore, the biological role of oxygen fluctuations observed in the more advanced stages of pregnancy and also after placenta development, although difficult to interpret, may serve for tissue-specific stem cell recruitment [55] and cell differentiation [56]. In some anatomic districts, for instance, hypoxemia is essential in preserving naïve stemness potential [57,58], or modulating cell differentiation [56]. This is particularly evident in the nervous central system, where hypoxia stimulates several processes, including cell survival, proliferation, catecholaminergic differentiation of isolated neural crest stem cells [59], mesencephalic precursor cells into neurons [60], or undifferentiated astrocytes to differentiate [61,62]. In other regions, instead, it is the increased oxygenation that drives cell differentiation, as in neural retinal tissue [63], pancreatic cells [64], keratinocytes [65], hepatocyte cell lines [66], or endothelial cells [67]. Therefore, it is realistic to speculate that the intrauterine environment, which physiologically becomes more hypoxic from mid-gestation to near-term and then less hypoxic until term, guarantees tissue-specific cellular differentiation depending on oxygen tension.

As shown here, the 34th week of gestation would represent the watershed between a hypoxic first phase, characterized by increased stem cell proliferation and the differentiation of specific cell populations, and a subsequent phase in which increasing oxygenation promotes the reduction in the stem cell pool and cell differentiation. This relationship between stemness and gestational age is well evident for endothelial precursor cells (EPCs), which are significantly more expressed in the umbilical cord blood of preterm newborns if compared with term neonates [68,69]. Interestingly, the cord blood of preterm infants contains a number of EPCs that grow as gestational age progresses: very low at 24–28 weeks, and then much more expressed at 28–35 weeks [70] or until 34 weeks [71]. Considering that the differentiation of endothelial cells is favored by the increase in oxygen levels [67], we hypothesize that the expansion of EPCs is synchronous with increasing hypoxemia and stops around 33–34th when oxygen increases, promoting the completion of the vasculature.

The topic of vasculogenesis is particularly intriguing, considering that in rodents, at variance with humans, the vascularization of many organs is immature at birth. This is particularly evident in the brain, where vascularization completes after birth [8,9], and even more so in the retina, which is avascular during intrauterine life but vascularizes after birth [10,11].

Although vasculature development depends on hypoxia-driven HIF and VEGF increases [38], in the retina, vessel appearance becomes evident a few days after birth, when oxygenation increases in concomitance with lowered levels of HIF and VEGF [72]. This observation suggests that vascularization occurs in two phases. In the first intrauterine phase, hypoxia induces astrocyte maturation and the production of VEGF [73], favoring, in turn, the recruitment of EPCs. Then, over the second phase, characterized by increased oxygenation and reductions in HIF and VEGF, EPCs can differentiate into endothelial cells. A similar immaturity is present in very preterm neonates, whose retina at birth is only partially vascularized and whose brain is incompletely vascularized, especially in the germinal matrix [74]. This immaturity is believed to predispose preterm infants to an elevated risk of developing intraventricular hemorrhage. However, this risk decreases for neonates born after 33–34 weeks of pregnancy or, in highly preterm newborns, after the first days of life, suggesting that a higher oxygenated environment induces endothelial cell maturation [74].

Considering that oxygen plays an important role in the modulation of the stem population and the processes of differentiation, knowledge of its fluctuations and the consequent biological effects opens new perspectives for the studies that, in the future, will try to artificially reproduce the benefits of intrauterine life. In this regard, studies aimed at the development of artificial placenta technology [75] cannot ignore the physiological levels of oxygenation, which must be respected. But even attempts to mimic the biological effects of intrauterine hypoxemia through the stimulation of β3-adrenoceptor agonists [76] will have to take into consideration that fetal hypoxemia increases only up to the 33–34th week.

### Limitations

This study has two main limitations:Umbilical cord samples represent a safe, non-invasive method to obtain valuable information regarding intrauterine conditions [77]. However, the type of delivery may affect the gas analytic values: fetal oxygenation of neonates born by vaginal delivery may, in fact, be altered by the engagement of the fetus through the birth canal or by the reduction in blood flow during uterine contractions [78], as well as by cesarean section, which can affect fetal well-being due to the cardiovascular effects of anesthesia, maternal position, and maternal ventilation [79,80]. Although the impact of delivery type on the gas test result is well known [13], in this study we did not find a different incidence of vaginal or cesarean delivery at different gestational ages, and therefore the blood samples were analyzed as a whole without further stratification.In this study, only umbilical cord blood samples from preterm infants were analyzed. The merging of data obtained from preterm and term newborns would have provided a better interpretation of the fluctuations in fetal oxygenation. However, the pooling of such data would not have been methodologically suitable, as umbilical cord blood samples in preterm neonates were collected within 60 sec after birth (“early cord clamping”), while samples in term neonates were collected approximately 60 sec after birth (“delayed cord clamping”) [13].

## 4. Materials and Methods

This retrospective study was performed at the University Hospital of Pisa, Italy, with around 1700 births per year. All the newborns born between January 1st 2016 and December 31st 2022 were eligible. Their umbilical cord blood samples were collected immediately after birth (early cord clamping). Samples collected from term newborns (≥ 37 weeks) or gestations with fetal or maternal intrapartum complications (i.e., fetuses with an operative vaginal delivery involving application of forceps or a vacuum extractor, abnormal intrapartum cardiotocography requiring emergency cesarean section, meconium-stained amniotic fluid, placental abruption, cord prolapse, chorioamnionitis, maternal sepsis, convulsions, hemorrhages, uterus rupture, snapped cord) were excluded from the study. Umbilical cord blood samples for which pH, base excess (BE), or both were not available and with acidosis at birth (pH ≤ 7.00 and/or BE ≤ −12 mmol/L) [81,82] were excluded from the analysis.

Values of umbilical (venous and arterial) parameters < or >3 SD from their respective means were individually evaluated and (i) rectified if probably mid-entered, (ii) maintained unchanged if considered plausible, and (iii) excluded if considered not plausible [14]. Results of cord blood gas that did not fulfill the following criteria: (i) arterial pH < the venous pH (by at least a difference of 0.022) and (ii) arterial pCO_2_ > the venous pCO_2_ (by at least a difference of 5.3 mmHg) were considered unreliable and excluded from the analysis [83].

Cord blood was collected, as recently published [13]. The samples, after their labeling and identification as venous or arterial, were analyzed as soon as possible, using an automatic blood gas analyzer (GEM^®^ Premier 4000, Instrumentation Laboratory, Lexington, MA, USA). The pH, pCO_2_, pO_2_, SaO_2_, lactate, and Hb were measured, whereas bicarbonate and base excess were calculated, respectively, from measured pH and pCO_2_ using the Henderson–Hasselbalch equation pH = 6.1 + log([HCO_3_^−^]/pCO_2_ × 0.03) [84] and the formula described by Siggaard-Anderson: (1 − 0.014 × Hb) × [HCO_3_^−^ − 24.8 + (1.43 × Hb + 7.7) × (pH − 7.4)] [85]. Oxygen content (mL/dL) was calculated using the formula [(1.36 × Hb (g/dL) × SaO_2_ (%)/100) + (0.0031 × pO_2_ (mmHg)], where 1.36 is the volume of O_2_ (mL) bound by a gram of Hb and 0.0031 is the solubility coefficient of O_2_ in the blood that represents the volume of O_2_ (mL) dissolved in 100 mL of plasma for each mmHg of O_2_ partial pressure. Fetal oxygen extraction was determined as the difference between umbilical venous and arterial blood oxygen contents divided by umbilical venous oxygen content. Fetal CO_2_ and lactate productions were calculated, respectively, as the difference between arterial blood CO_2_ or lactate and venous contents, divided by the respective venous content.

### Statistical Analysis

Categorical data were described with the absolute and relative (%) frequency, and continuous data were summarized with the mean and standard deviation. Gestational age was compared with different parameters of interest (umbilical venous and arterial oxygen levels, fetal oxygen extraction, Hb, oxygen content, pH, bicarbonate, base excess, CO_2_, and lactate) using regression models to identify the best model between the linear and the quadratic ones. Furthermore, the curves and the equations of the best models were calculated, and the equation of the statistically significant model (linear or quadratic) was indicated at the top of each figure. When the most representative test (with a lower *p*) was a linear regression, the test revealed that the parameter had a linear trend (decreasing or increasing) with time. Conversely, when the most representative test (with a lower *p*) was a quadratic regression, the test revealed that the parameter had a biphasic trend with time: If the quadratic model was increasing, the trend reached a maximum value and was subsequently decreasing, or if the quadratic model was decreasing, the trend reached a minimum value and was subsequently increasing. Therefore, this test suggested when, approximately, the curve reversed.

Once the data were stratified according to weeks of pregnancy (23–26^+6^, 27–28^+6^, 29–30^+6^, 31–33^+6^, 34–35^+6^, and 36–36^+6^ weeks of gestation), a nonparametric test without Bonferroni correction was employed to evaluate the differences between the groups (Appendix A). Significance was set at 0.05, and all analyses were performed by SPSS v.28 technology (IBM Corp. Released 2021. IBM SPSS Statistics for Windows, Version 28.0. IBM Corp: Armonk, NY, USA).

## 5. Conclusions

The present study confirms that oxygenation during fetal life undergoes important fluctuations and integrates the recently published study on oxygenation status during the last weeks of pregnancy [13]. The combination of these studies suggests that from the 23rd week onwards, the fetus becomes progressively more hypoxemic, but this condition reverses around the 33–34th week when oxygen levels progressively increase until the end of the pregnancy.

## Figures and Tables

**Figure 1 ijms-24-12487-f001:**
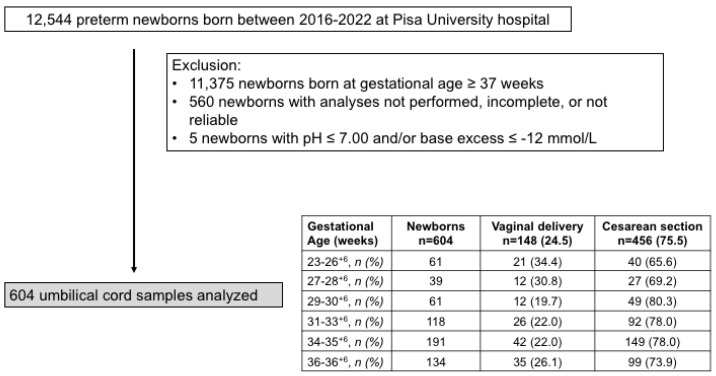
Flow chart illustrating patient enrollment of this retrospective observational cohort study.

**Figure 2 ijms-24-12487-f002:**
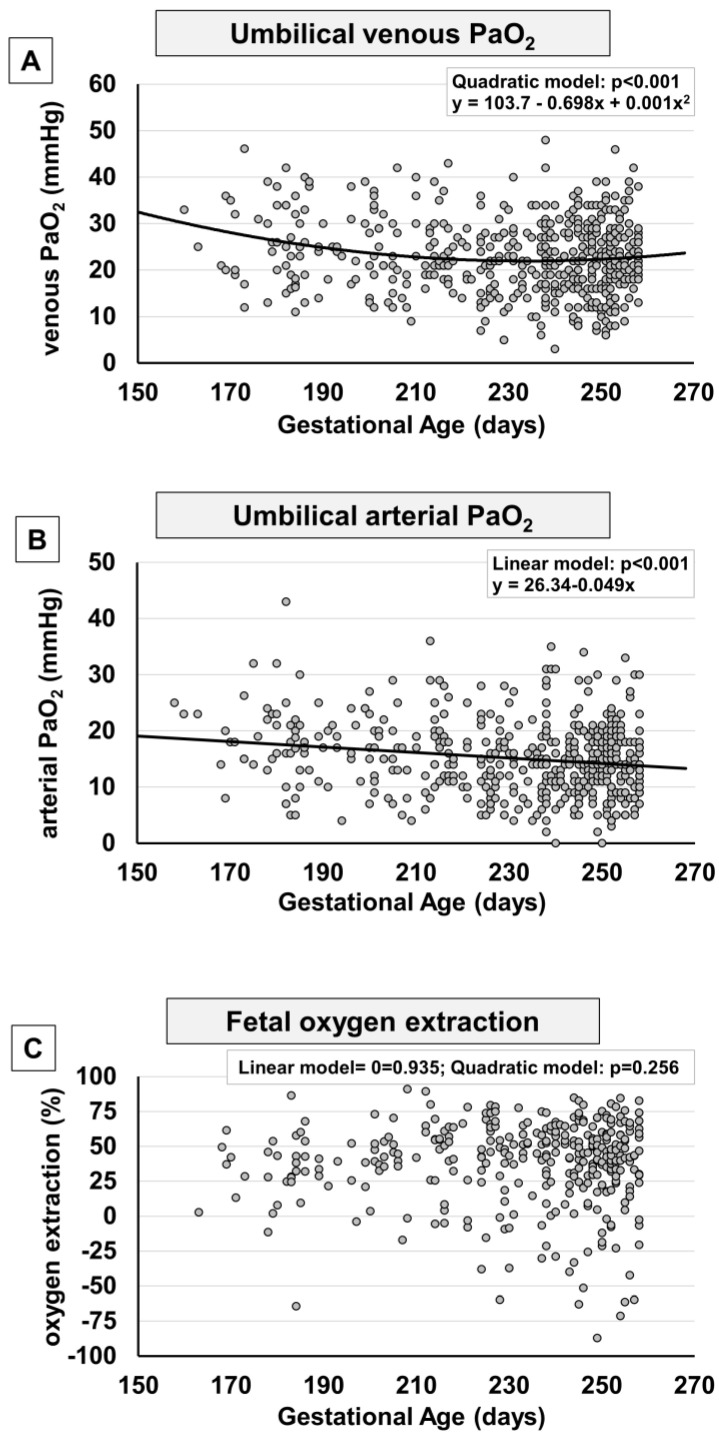
Scatter plots with regression lines representing umbilical cord oxygenation status [venous pO_2_, panel (**A**) (*n* = 532); arterial pO_2_, panel (**B**) (*n* = 500); fetal oxygen extraction, panel (**C**) (*n* = 399)] of the whole study population. The equation of the statistically significant model (linear or quadratic) is indicated at the top of each figure.

**Figure 3 ijms-24-12487-f003:**
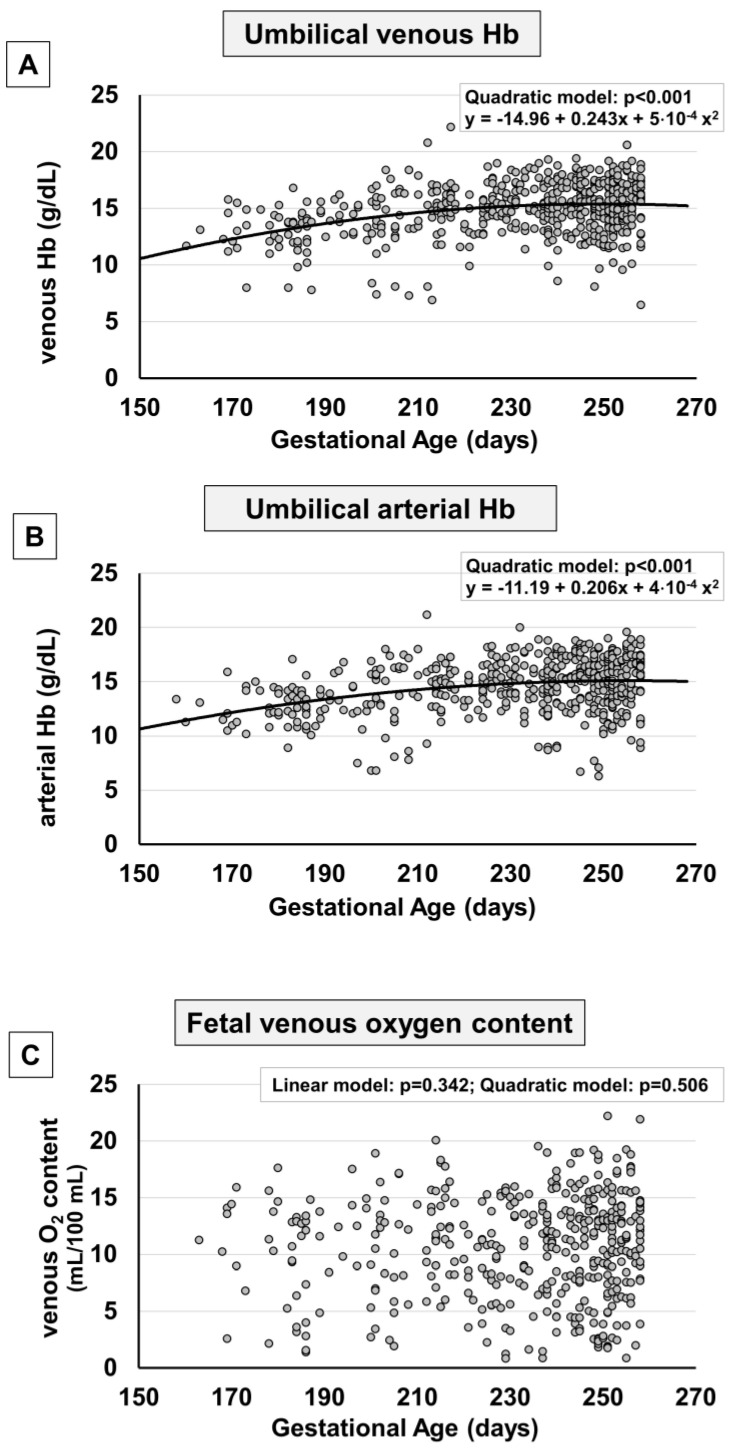
Scatter plots with regression lines representing umbilical cord Hb and oxygen content [venous Hb, panel (**A**) (*n* = 566); arterial Hb, panel (**B**) (*n* = 534); fetal venous oxygen content, panel (**C**) (*n* = 427)] of the whole study population. The equation of the statistically significant model (linear or quadratic) is indicated at the top of each figure.

**Figure 4 ijms-24-12487-f004:**
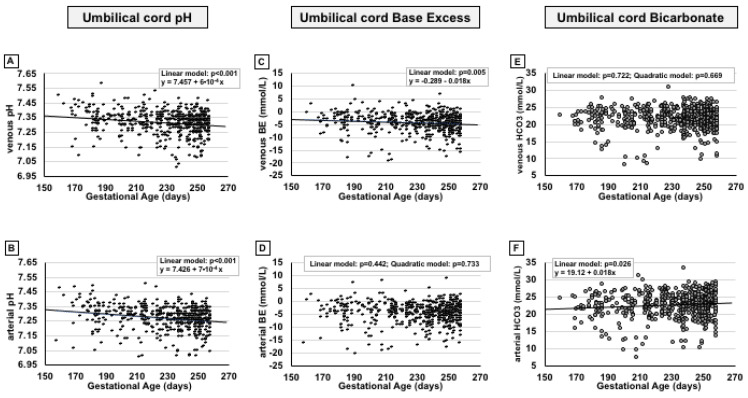
Scatter plots with regression lines representing umbilical cord pH, base excess, and bicarbonate [venous pH, panel (**A**) (*n* = 584); arterial pH, panel (**B**) (*n* = 552); venous base excess, panel (**C**) (*n* = 582); arterial base excess, panel (**D**) (*n* = 554); venous bicarbonate, panel (**E**) (*n* = 573); arterial bicarbonate, panel (**F**) (*n* = 541)] of the whole study population. The equation of the statistically significant model (linear or quadratic) is indicated at the top of each figure.

**Figure 5 ijms-24-12487-f005:**
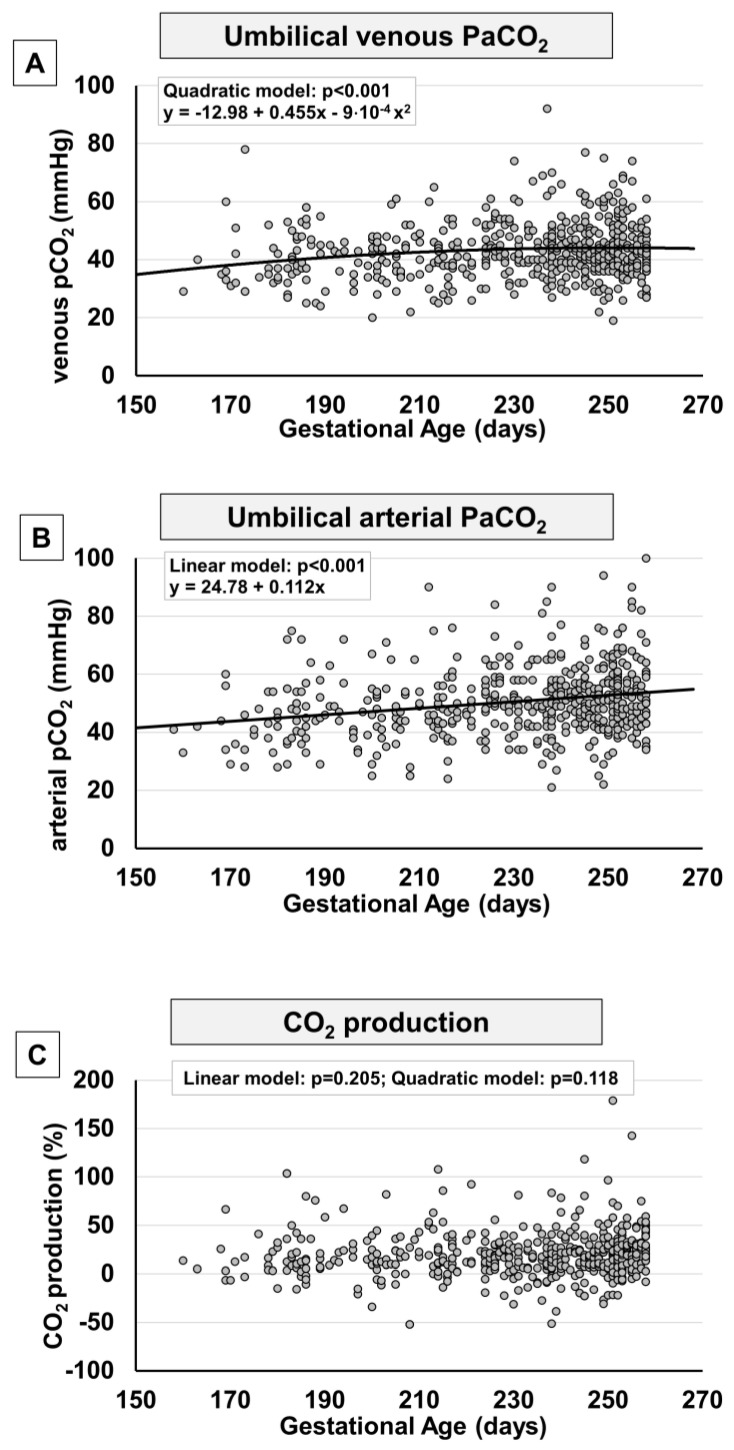
Scatter plots with regression lines representing umbilical cord carbon dioxide levels [venous pCO_2_, panel (**A**) (*n* = 580); arterial pCO_2_, panel (**B**) (*n* = 548); fetal CO_2_ production, panel (**C**) (*n* = 527)]. The equation of the statistically significant model (linear or quadratic) is indicated at the top of each figure.

**Figure 6 ijms-24-12487-f006:**
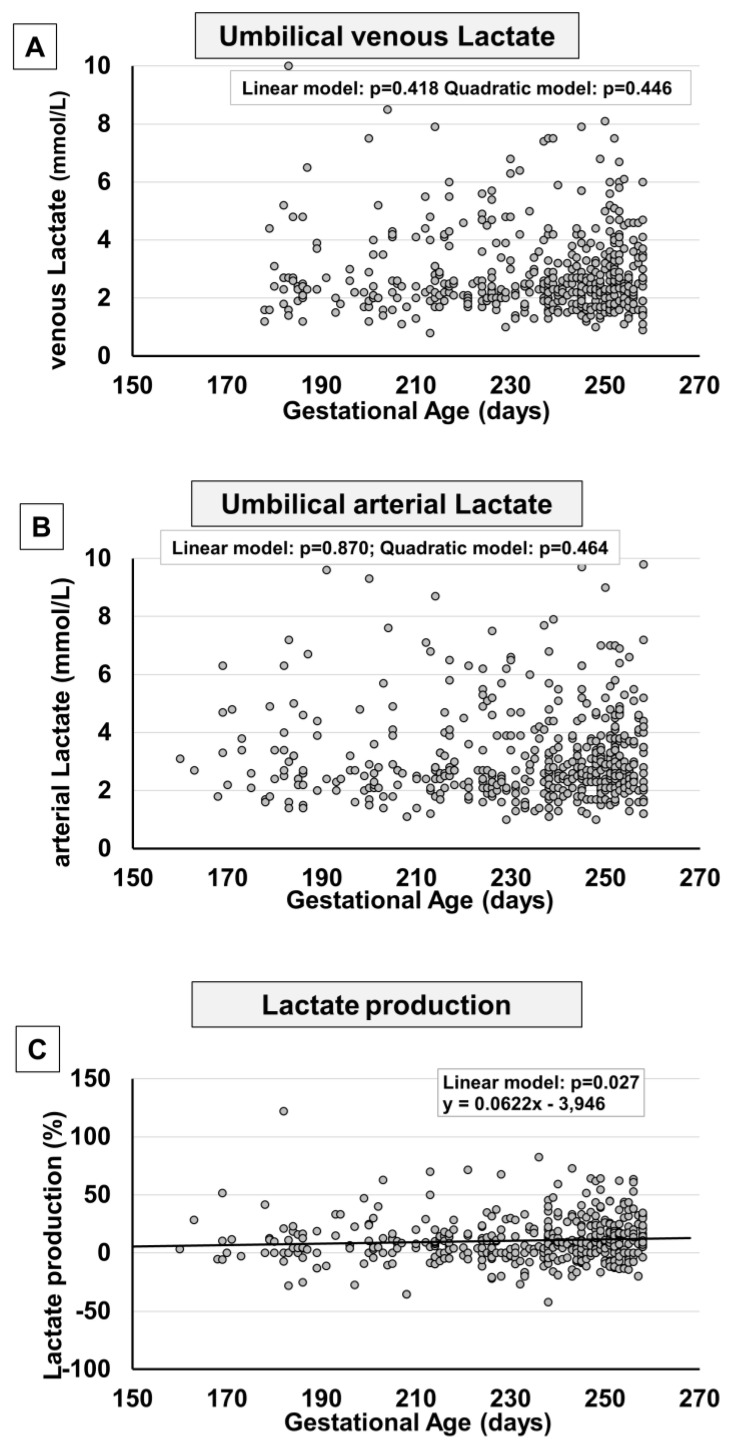
Scatter plots with regression lines representing umbilical cord lactate levels [venous lactate, panel (**A**) (*n* = 513); arterial lactate, panel (**B**) (*n* = 481); fetal lactate production, panel (**C**) (*n* = 465)]. The equation of the statistically significant model (linear or quadratic) is indicated at the top of each figure.

**Table 1 ijms-24-12487-t001:** Umbilical cord blood gas analysis in all enrolled preterm newborns and separately analyzed by the type of delivery.

	All Preterm Newborns*n* = 604	Vaginal Delivery*n* = 148	Cesarean Section*n* = 456	*p* Value
GA, days, mean (SD)	230 (24)	228 (28)	231 (23)	0.127
Birth weight, g, mean (SD)	1855 (687)	1926 (728)	1832 (672)	0.155
Male, n (%)	310 (51.2)	80 (54.0)	230 (50.4)	0.474
Apgar Score at 5 min, mean (SD)	7.6 (1.2)	7.6 (1.4)	7.7 (1.1)	0.781
Umbilical venous cord sampling
pH, mean (SD)	7.315 (0.08)	7.334 (0.08)	7.309 (0.08)	<0.001
pCO_2_, mmHg, mean (SD)	43.2 (9.4)	39.9 (9.5)	44.3 (9.2)	<0.001
pO_2_, mmHg, mean (SD)	22.8 (7.9)	25.3 (8.3)	22.0 (7.6)	<0.001
Bicarbonate, mmol/L, mean (SD)	21.7 (3.1)	20.9 (2.7)	22.0 (3.1)	<0.001
BE(B), mmol/L, mean (SD)	−4.3 (3.6)	−4.6 (3.5)	−4.3 (3.6)	0.250
Lactate, mmol/L, mean (SD)	2.8 (1.4)	3.3 (1.4)	2.7 (1.3)	<0.001
Hemoglobin, g/dL, mean (SD)	14.96 (2.3)	15.5 (1.9)	14.8 (2.4)	0.001
SaO_2_, %, mean (SD)	52.9 (22.1)	57.4 (21.4)	51.3 (22.2)	0.011
Umbilical arterial cord sampling
pH, mean (SD)	7.267 (0.08)	7.274 (0.09)	7.278 (0.08)	0.327
pCO_2_, mmHg, mean (SD)	50.7 (11.4)	49.7 (11.8)	51.0 (11.3)	0.243
pO_2_, mmHg, mean (SD)	15.2 (6.9)	17.6 (6.7)	14.4 (6.8)	<0.001
Bicarbonate, mmol/L, mean (SD)	22.7 (3.8)	22.3 (3.2)	22.8 (3.9)	0.227
BE(B), mmol/L, mean (SD)	−4.6 (4.1)	−4.8 (3.7)	−4.5 (4.3)	0.414
Lactate, mmol/L, mean (SD)	3.1 (1.5)	3.7 (1.6)	2.9 (1.5)	<0.001
Hemoglobin, g/dL, mean (SD)	14.96 (2.3)	15.3 (2.1)	14.4 (2.5)	<0.001
SaO_2_, %, mean (SD)	33.0 (19.2)	40.4 (20.0)	30.0 (18.2)	<0.001

Veno-arterial O_2_ difference, mmHg, mean (SD)	7.7 (6.4)	7.5 (7.1)	7.8 (6.2)	0.665
Fetal oxygen extraction, %, mean (SD)	35.7 (35.8)	27.3 (32.0)	38.6 (36.8)	0.007
GA = gestational age; pO_2_ = partial pressure of oxygen; pCO_2_ = partial pressure of carbon dioxide; BE = base excess.

## Data Availability

The datasets generated during and/or analyzed during the current study are available from the corresponding author on reasonable request.

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
