# Peer review of "Fetal Oxygenation from the 23rd to the 36th Week of Gestation Evaluated through the Umbilical Cord Blood Gas Analysis"

_ijms, 2023, doi:10.3390/ijms241512487_

Round 1

Reviewer 1 Report

Comments and Suggestions for Authors

Comments to Authors

Thank you for the opportunity to review “Fetal oxygenation from the 23rd to the 36th week of gestation evaluated through the umbilical cord blood gas analysis”. The authors reported umbilical-cord blood gas analysis at delivery from the 23rd to the 36th weeks of gestation. The manuscript was written well but I have the following minor concerns regarding this paper.

Methods part

Was this study prospective cohort? The authors should describe study design in the methods part.

Figure 1

The range of gestational weeks written in Figure 1 was between 22 to 36 weeks. Was the range from the 23rd to the 36th weeks of gestation?

Table 1

Was Apgar score at 5 min or 1 min after birth?

Results part

Why was the fetal oxygen extraction obtained in only 399 cases while pO2 and pCO2 were obtained in more cases? If fetal oxygen extraction was calculated using only umbilical venous oxygen content and arterial blood oxygen contents, fetal oxygen extraction should be obtained in more cases.

Author Response

To the Editor of
International Journal of Molecular Sciences

Pisa, August 02, 2023

Dear Editor,

            we hereby submit the revised version of the paper “Fetal oxygenation from the 23rd to the 36th week of gestation evaluated through the umbilical cord blood gas analysis” by Luca Filippi et al., for possible publication as a Research Article within the Special Issue “Physiology and Pathophysiology of Placenta" of IJMS.

The manuscript has been thoroughly revised taking into account the criticisms and the suggestions of the Reviewers.

The changes in the text are written in red. 

Reply to Reviewer 1

Reviewer 1 wrote: Methods part “Was this study prospective cohort? The authors should describe study design in the methods part”.

According to this observation (and with observation of the reviewer #2) we have specified in the Method Section that this study analyzed retrospectively the umbilical-cord blood samples collected from preterm neonates born between January 1st, 2016, and December 31st, 2022 (“This retrospective study…”; line 423). A similar declaration has been added in the Abstract (line 16).

Reviewer 1 wrote: Figure 1 “The range of gestational weeks written in Figure 1 was between 22 to 36 weeks. Was the range from the 23rd to the 36th weeks of gestation?”

Thank you for this remark. Our population is composed of patients with gestational age between the 23rd and the 36th week. Therefore, we have corrected the Figure 1 and the text.

Reviewer 1 wrote: Table 1. “Was Apgar score at 5 min or 1 min after birth?”

The Apgar score was at 5 min. We have corrected Table 1.

Reviewer 1 wrote: Results part “Why was the fetal oxygen extraction obtained in only 399 cases while pO2 and pCO2 were obtained in more cases? If fetal oxygen extraction was calculated using only umbilical venous oxygen content and arterial blood oxygen contents, fetal oxygen extraction should be obtained in more cases”.

Fetal oxygen extraction was obtained by calculating the difference between the umbilical venous and the arterial blood oxygen contents divided by the umbilical venous oxygen content. We had data on umbilical venous oxygen content from 427 neonates, and data on umbilical arterial oxygen content from 415 neonates; however, only 399 neonates had both values, and therefore we could calculate the fetal oxygen extraction in 399 neonates.

Reply to Reviewer 2

Reviewer 2 wrote: “In the methods provide more detail on statistical tools used in the study (what type of tests/regression analyses etc.). In supplementary figures authors present multiple comparison – please confirm if there were corrections for multiple comparison”.

The values of the different parameters of interest (umbilical venous and arterial oxygen levels, fetal oxygen extraction, Hb, oxygen content, pH, bicarbonate, base excess, CO2, and lactate) were plotted as a function of gestational age. Then, these values were subjected to linear and quadratic regression tests. The test that best represented (with a lower p) the trend of the parameter is shown in the Figures. When the most representative test (with a lower p) was a linear regression, the test revealed that the parameter had a linear trend (decreasing or increasing) with time. Conversely, when the most representative test (with a lower p) was a quadratic regression, the test revealed that the parameter had a biphasic trend with time: if the quadratic model is increasing, the trend reaches a maximum value and is subsequently decreasing, or if the quadratic model is decreasing, the trend reaches a minimum value and is subsequently increasing. Therefore, this test suggests when, approximately, the curve reverses. In the text, more details are provided regarding statistical strategy and are highlighted in red (lines 458-472).

Moreover, in the text, we specified: “When data were stratified according to weeks of pregnancy (23-26+6, 27-28+6, 29-30+6, 31-33+6, 34-35+6, and 36-36+6 weeks of gestation) a nonparametric test without Bonferroni correction was employed to evaluate the differences between the groups (Supplementary Figures)” (lines 472-475).

Reviewer 2 wrote: “Please, provide more specific statistical information in the description of results. The current description of results basically lacks any reference to statistical analysis. It is no clear which and if any changes are statistically and clinically important (some examples of imprecise phrasing – but the comment is pertinent to all results: “seems to decrease” L152, “opposing tendency if compared with the trend of oxygen, with a progressive surge both in venous (…) arterial samples” L149-150; "This increase appears to be progressive at least until the group with gestational age 31-33+6 weeks” L150-151)”.

We completely agree with the observation of the reviewer, and the text was amended accordingly.  The sentence “pO2 seems to increase” at line 114 was changed into “pO2 increases” (line 114), the sentence “an opposing tendence” into “an opposing trend” (line 160), the sentence “This increase appears to be progressive” into “This increase is progressive” (line 162), the sentence “pCO2 seems to decrease” into “pCO2 decreases” (line 164), the sentence “which seems to represent” into “which represents” (line 296).

Reviewer 2 wrote: “Please, provide more detail on statistical results presented in Figs 2-6 (which show some statistical information on the figures themselves). This should be presented both in the text of the results and figure captions”.

According to the reviewer’s suggestion, we added in the Figures’ caption the sentence “The equation of the statistically significant model (linear or quadratic) is indicated at the top of each figure”. In the text, we added additional comments regarding the statistical contribution of results. For example, we added in lines 110-112 “… Values from umbilical venous samples revealed a biphasic trend of oxygen level, as suggested by the significance of the quadratic regression: pO2 levels progressively decrease …

Reviewer 2 wrote: “Caption of Fig. 1 indicates that this is a retrospective study. Please, clearly state this in the abstract and in the methods”.

According to the reviewer’s observation, we stated in the Abstract (line 16) and in the Method Section (line 423) that this is a retrospective study.

Reviewer 2 wrote: “Since there are significant differences of umbilical cord blood between vaginal vs caesarean delivery, do the authors expect differences in the temporal pattern of oxygen, carbon dioxide, acid-base balance in these two groups?”

We do not expect any differences in temporal patterns between neonates born by vaginal and caesarean delivery. We observed significant differences in absolute values of oxygen, carbon dioxide, and acid-base values between infants born by vaginal or cesarean delivery due to the effect of delivery type, as recently observed in term newborns. But there is no reason to imagine that the dynamics of these values as the weeks pass during intrauterine life are influenced by the type of delivery. Recently, in a study performed in term neonates, we have recently demonstrated that the type of delivery influenced the absolute values of umbilical blood gas analysis, but not their dynamic over the weeks (doi: 10.3389/fped.2023.1140021).

Minor points:

Reviewer 2 wrote: “If possible, I suggest providing larger figures for the final production of the MS. Current version is quite small (especially font)”.

According to your observation, we provided larger Figures and we increased the font to all images

Reviewer 2 wrote: “Last sentence of limitations (L391-393): “However, the pooling of such data would not have been methodologically suitable, as umbilical-cord blood samples in neonates born at term are usually performed by blood collected approximately 60 sec after birth [13].” – I suggest adding “early cord clamping” (within 60 sec after birth) in this study, and “delayed cord clamping” in the study no 13”.

According to the reviewer’s observation, we wrote: “In this study, only umbilical-cord blood samples from preterm infants were analyzed. … the pooling of such data would not have been methodologically suitable, as umbilical-cord blood samples in preterm neonates were collected within 60 sec after birth (“early cord clamping”) while samples of term neonates were collected approximately 60 sec after birth (“delayed cord clamping”)” (lines 407-413).

For negotiations concerning the manuscript please contact:

Luca Filippi, MD, PhD, Department of Clinical and Experimental Medicine, University of Pisa, Via Roma, 67 I-56126 Pisa, Italy. E-mail: luca.filippi@unipi.it - Tel: +39-50-993677

Thank you for your attention to our paper.

                        Yours sincerely,

Prof. Luca Filippi, MD, PhD

Reviewer 2 Report

Comments and Suggestions for Authors

The authors in their manuscript (MS) entitled “Fetal oxygenation from the 23rd to the 36th week of gestation evaluated through the umbilical cord blood gas analysis” present retrospective observational study in human preterm newborns that evaluated oxygenation of the umibilical blood as a proxy of fetal blood oxygenation. The authors show a temporal pattern of oxygen partial pressure and content in the umbilical blood immediately after the birth dependent on the gestational week.

The MS is generally well written and concise, with references to relevant and current literature. In my view the results are interesting. The main flaw of the MS is imprecise description of statistics and lack of statistical detail in the presentation of results.

Major points:

1. In the methods provide more detail on statistical tools used in the study (what type of tests/regression analyses etc.). In supplementary figures authors present multiple comparison – please confirm if there were corrections for multiple comparison.

2. Please, provide more specific statistical information in the description of results. The current description of results basically lacks any reference to statistical analysis. It is no clear which and if any changes are statistically and clinically important (some examples of imprecise phrasing – but the comment is pertinent to all results: “seems to decrease” L152, “opposing tendency if compared with the trend of oxygen, with a progressive surge both in venous (…) arterial samples” L149-150; "This increase appears to be progressive at least until the group with gestational age 31-33+6 weeks” L150-151).

3. Please, provide more detail on statistical results presented in Figs 2-6 (which show some statistical information on the figures themselves). This should be presented both in the text of the results and figure captions.

4. Caption of Fig. 1 indicates that this is a retrospective study. Please, clearly state this in the abstract and in the methods.

5. Since there are significant differences of umbilical cord blood between  vaginal vs caesarean delivery, do the authors expect differences in the temporal pattern of oxygen, carbon dioxide, acid-base balance in these two groups?

Minor points:

1. If possible, I suggest providing larger figures for the final production of the MS. Current version is quite small (especially font).

2. Last sentence of limitations (L391-393): “However, the pooling  of such data would not have been methodologically suitable, as umbilical-cord blood samples in neonates born at term are usually performed by blood collected approximately 60 sec after birth [13].” – I suggest adding “early cord clamping” (within 60 sec after birth) in this study, and “delayed cord clamping” in the study no 13.

Author Response

(The authors gave the same response as above.)

Round 2

Reviewer 2 Report

Comments and Suggestions for Authors

Thank you for addressing my all concerns.

In my view the MS is significantly improved.